# Gradients of Counterfactuals

**Mukund Sundararajan, Ankur Taly & Qiqi Yan**
Google Inc.
Mountain View, CA 94043, USA
{mukunds,ataly,qiqiyan}@google.com

## Abstract

Gradients have been used to quantify feature importance in machine learning models. Unfortunately, in nonlinear deep networks, not only individual neurons but also the whole network can saturate, and as a result an important input feature can have a tiny gradient. We study various networks, and observe that this phenomena is indeed widespread, across many inputs.

We propose to examine *interior gradients*, which are gradients of counterfactual inputs constructed by scaling down the original input. We apply our method to the GoogleNet architecture for object recognition in images, as well as a ligand-based virtual screening network with categorical features and an LSTM based language model for the Penn Treebank dataset. We visualize how interior gradients better capture feature importance. Furthermore, interior gradients are applicable to a wide variety of deep networks, and have the *attribution* property that the feature importance scores sum to the the prediction score.

Best of all, interior gradients can be computed *just as easily as* gradients. In contrast, previous methods are complex to implement, which hinders practical adoption.

## 1 Introduction

Practitioners of machine learning regularly inspect the coefficients of linear models as a measure of feature importance. This process allows them to understand and debug these models. The natural analog of these coefficients for deep models are the gradients of the prediction score with respect to the input. For linear models, the gradient of an input feature is equal to its coefficient. For deep nonlinear models, the gradient can be thought of as a local linear approximation (Simonyan et al. (2013)). Unfortunately, (see the next section), the network can saturate and as a result an important input feature can have a tiny gradient.

While there has been other work (see Section 2.10) to address this problem, these techniques involve instrumenting the network. This instrumentation currently involves significant developer effort because they are not primitive operations in standard machine learning libraries. Besides, these techniques are not simple to understand—they invert the operation of the network in different ways, and have their own peculiarities—for instance, the feature importances are not invariant over networks that compute the exact same function (see Figure 14).

In contrast, the method we propose builds on the very familiar, primitive concept of the gradient—all it involves is inspecting the gradients of a few carefully chosen counterfactual inputs that are scaled versions of the initial input. This allows anyone who knows how to extract gradients—presumably even novice practitioners that are not very familiar with the network's implementation—to *debug* the network. Ultimately, this seems essential to ensuring that deep networks perform predictably when deployed.

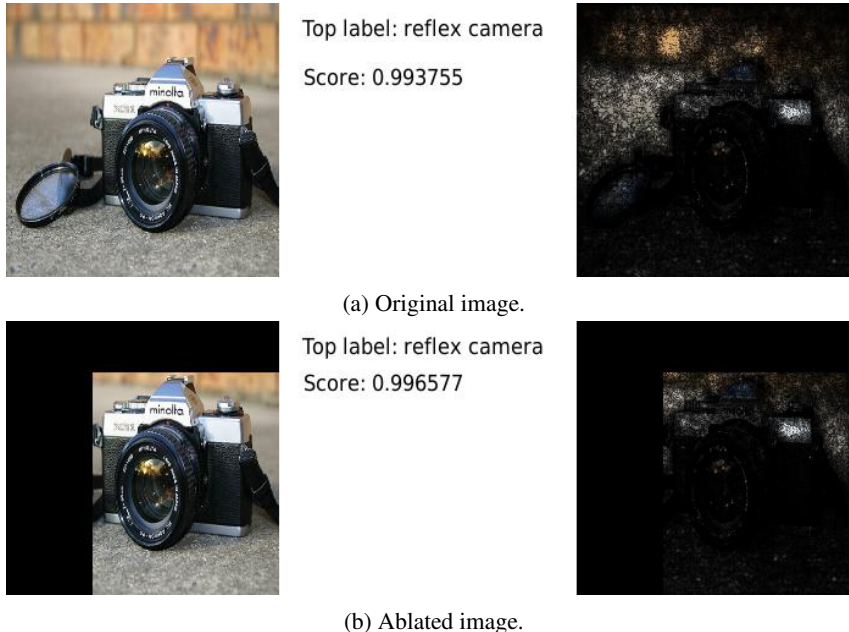

(a) Original image.

(b) Ablated image.

Figure 1: Pixel importance using gradients at the image.

## 2   OUR TECHNIQUE

### 2.1   GRADIENTS DO NOT REFLECT FEATURE IMPORTANCE

Let us start by investigating the performance of gradients as a measure of feature importance. We use an object recognition network built using the GoogleNet architecture (Szegedy et al. (2014)) as a running example; we refer to this network by its codename Inception. (We present applications of our techniques to other networks in Section 3.) The network has been trained on the ImageNet object recognition dataset (Russakovsky et al. (2015)). It is is 22 layers deep with a softmax layer on top for classifying images into one of the 1000 ImageNet object classes. The input to the network is a $224 \times 224$ sized RGB image.

Before evaluating the use of gradients for feature importance, we introduce some basic notation that is used throughout the paper.

We represent a $224 \times 224$ sized RGB image as a vector in $R^{224 \times 224 \times 3}$. Let $\mathsf{Incp}^L : R^{224 \times 224 \times 3} \rightarrow [0, 1]$ be the function represented by the Inception network that computes the softmax score for the object class labeled $L$. Let $\bigtriangledown \mathsf{Incp}^L(\mathsf{img})$ be the gradients of $\mathsf{Incp}^L$ at the input image img. Thus, the vector $\bigtriangledown \mathsf{Incp}^L(\mathsf{img})$ is the same size as the image and lies in $R^{224 \times 224 \times 3}$. As a shorthand, we write $\bigtriangledown \mathsf{Incp}^L_{i,j,c}(\mathsf{img})$ for the gradient of a specific pixel $(i, j)$ and color channel $c \in \{R, G, B\}$.

We compute the gradients of $\mathsf{Incp}^L$ (with respect to the image) for the highest-scoring object class, and then aggregate the gradients $\bigtriangledown \mathsf{Incp}^L(\mathsf{img})$ along the color dimension to obtain pixel importance scores.[1]

$$\forall i, j : \mathcal{P}^L_{i,j}(\mathsf{img}) ::= \Sigma_{c \in \{R,G,B\}} |\bigtriangledown \mathsf{Incp}^L_{i,j,c}(\mathsf{img})| \tag{1}$$

Next, we visualize pixel importance scores by scaling the intensities of the pixels in the original image in proportion to their respective scores; thus, higher the score brighter would be the pixel. Figure 1a shows a visualization for an image for which the highest scoring object class is "reflex camera" with a softmax score of $0.9938$.

---

[1] These pixel importance scores are similar to the gradient-based saliency map defined by Simonyan et al. (2013) with the difference being in how the gradients are aggregated along the color channel.

Intuitively, one would expect the the high gradient pixels for this classification to be ones falling on the camera or those providing useful context for the classification (e.g., the lens cap). However, most of the highlighted pixels seem to be on the left or above the camera, which to a human seem not essential to the prediction. This could either mean that (1) the highlighted pixels are somehow important for the internal computation performed by the Inception network, or (2) gradients of the image fail to appropriately quantify pixel importance.

Let us consider hypothesis (1). In order to test it we ablate parts of the image on the left and above the camera (by zeroing out the pixel intensities) and run the ablated image through the Inception network. See Figure 1b. The top predicted category still remains "reflex camera" with a softmax score of 0.9966 — slightly higher than before. This indicates that the ablated portions are indeed irrelevant to the classification. On computing gradients of the ablated image, we still find that most of the high gradient pixels lie outside of the camera. This suggests that for this image, it is in fact hypothesis (2) that holds true. Upon studying more images (see Figure 4), we find that the gradients often fail to highlight the relevant pixels for the predicted object label.

## 2.2 SATURATION

In theory, it is easy to see that the gradients may not reflect feature importance if the prediction function flattens in the vicinity of the input, or equivalently, the gradient of the prediction function with respect to the input is tiny in the vicinity of the input vector. This is what we call *saturation*, which has also been reported in previous work (Shrikumar et al. (2016), Glorot & Bengio (2010)).

We analyze how widespread saturation is in the Inception network by inspecting the behavior of the network on **counterfactual images** obtained by uniformly scaling pixel intensities from zero to their values in an actual image. Formally, given an input image img $\in R^{224 \times 224 \times 3}$, the set of counterfactual images is

$$\{\alpha \, \textsf{img} \mid 0 \leq \alpha \leq 1\} \tag{2}$$

Figure 2a shows the trend in the softmax output of the highest scoring class, for thirty randomly chosen images form the ImageNet dataset. More specifically, for each image img, it shows the trend in $\textsf{Incp}^L(\alpha \, \textsf{img})$ as $\alpha$ varies from zero to one with $L$ being the label of highest scoring object class for img. It is easy to see that the trend flattens (saturates) for all images $\alpha$ increases. Notice that saturation is present even for images whose final score is significantly below 1.0. Moreover, for a majority of images, saturation happens quite soon when $\alpha = 0.2$.

One may argue that since the output of the Inception network is the result of applying the softmax function to a vector of activation values, the saturation is expected due to the squashing property of the softmax function. However, as shown in Figure 2b, we find that even the pre-softmax activation scores for the highest scoring class saturate.

In fact, to our surprise, we found that the saturation is inherently present in the Inception network and the outputs of the intermediate layers also saturate. We plot the distance between the intermediate layer neuron activations for a scaled down input image and the actual input image with respect to the scaling parameter, and find that the trend flattens. Due to lack of space, we provide these plots in Figure 12 in the appendix.

It is quite clear from these plots that saturation is widespread across images in the Inception network, and there is a lot more activity in the network for counterfactual images at relatively low values of the scaling parameter $\alpha$. This observation forms the basis of our technique for quantifying feature importance.

Note that it is well known that the saturation of gradients prevent the model from converging to a good quality minima (Glorot & Bengio (2010)). So one may expect good quality models to not have saturation and hence for the (final) gradients to convey feature importance. Clearly, our observations on the Inception model show that this is not the case. It has good prediction accuracy, but also exhibits saturation (see Figure 2). Our hypothesis is that the gradients of important features are *not* saturated early in the training process. The gradients only saturate *after* the features have been learned adequately, i.e., the input is far away from the decision boundary.

## 2.3 INTERIOR GRADIENTS

We study the importance of input features in a prediction made for an input by examining the gradients of the counterfactuals obtained by scaling the input; we call this set of gradients **interior gradients**.

While the method of examining gradients of counterfactual inputs is broadly applicable to a wide range of networks, we first explain it in the context of Inception. Here, the counterfactual image inputs we consider are obtained by uniformly scaling pixel intensities from zero to their values in the actual image (this is the same set of counterfactuals that was used to study saturation). The interior gradients are the gradients of these images.

$$\mathsf{InteriorGrads(img)} ::= \{\triangledown\mathsf{Incp}(\alpha\,\mathsf{img}) \mid 0 \leq \alpha \leq 1\} \tag{3}$$

These interior gradients explore the behavior of the network along the entire scaling curve depicted in Figure 2a, rather than at a specific point. We can aggregate the interior gradients along the color dimension to obtain interior pixel importance scores using equation 1.

$$\mathsf{InteriorPixelImportance(img)} ::= \{\mathcal{P}(\alpha\,\mathsf{img}) \mid 0 \leq \alpha \leq 1\} \tag{4}$$

We individually visualize the pixel importance scores for each scaling parameter $\alpha$ by scaling the intensities of the pixels in the actual image in proportion to their scores. The visualizations show how the importance of each pixel evolves as we scale the image, with the last visualization being identical to one generated by gradients at the actual image. In this regard, the interior gradients offer strictly more insight into pixel importance than just the gradients at the actual image.

Figure 3 shows the visualizations for the "reflex camera" image from Figure 1a for various values of the scaling parameter $\alpha$. The plot in the top right corner shows the trend in the absolute magnitude of the average pixel importance score. The magnitude is significantly larger at lower values of $\alpha$ and nearly zero at higher values — the latter is a consequence of saturation. Note that each visualization is only indicative of the relative distribution of the importance scores across pixels and not the absolute magnitude of the scores, i.e., the later snapshots are responsible for tiny increases in the scores as the chart in the top right depicts.

The visualizations show that at lower values of $\alpha$, the pixels that lie on the camera are most important, and as $\alpha$ increases, the region above the camera gains importance. Given the high magnitude of gradients at lower values of $\alpha$, we consider those gradients to be the primary drivers of the final prediction score. They are more indicative of feature importance in the prediction compared to the gradients at the actual image (i.e., when $\alpha = 1$).

The visualizations of the interior pixel gradients can also be viewed together as a single animation that chains the visualizations in sequence of the scaling parameter. This animation offers a concise yet complete summary of how pixel importance moves around the image as the scaling parameter increase from zero to one.

**Rationale.** While measuring saturation via counterfactuals seems natural, using them for quantifying feature importance deserves some discussion. The first thing one may try to identify feature importance is to examine the deep network like one would with human authored code. This seems hard; just as deep networks employ distributed representations (such as embeddings), they perform convoluted (pun intended) distributed reasoning. So instead, we choose to probe the network with several counterfactual inputs (related to the input at hand), hoping to trigger all the internal workings of the network. This process would help summarize the effect of the network on the protagonist input; the assumption being that the input is human understandable. Naturally, it helps to work with gradients in this process as via back propagation, they induce an aggregate view over the function computed by the neurons.

Interior gradients use counterfactual inputs to artifactually induce a procedure on how the networks attention moves across the image as it compute the final prediction score. From the animation, we gather that the network focuses on strong and distinctive patterns in the image at lower values of the scaling parameter, and subtle and weak patterns in the image at higher values. Thus, we speculate that the network's computation can be loosely abstracted by a procedure that first recognize distinctive features of the image to make an initial prediction, and then fine tunes (these are small score jumps as the chart in Figure 3 shows) the prediction using weaker patterns in the image.

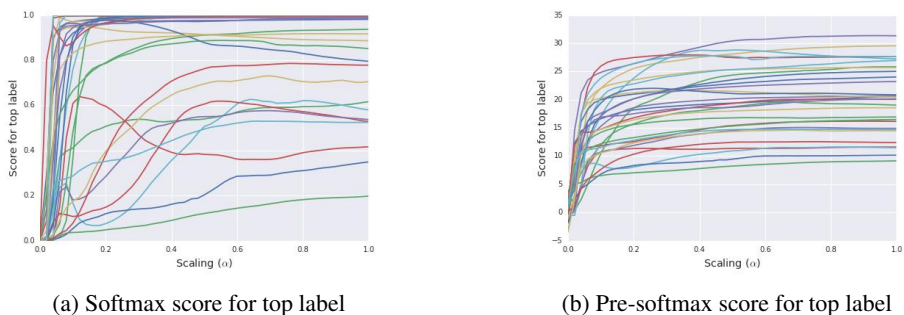

(a) Softmax score for top label (b) Pre-softmax score for top label

Figure 2: Saturation in Inception

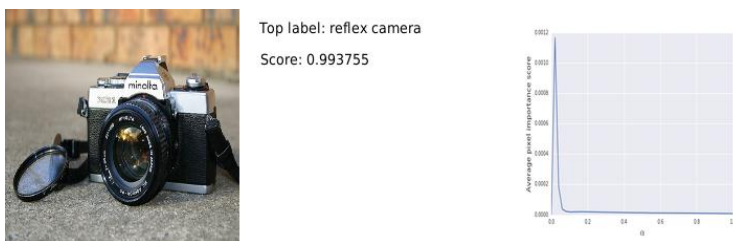

Input image and trend of the pixel importance scores obtained from interior gradients.

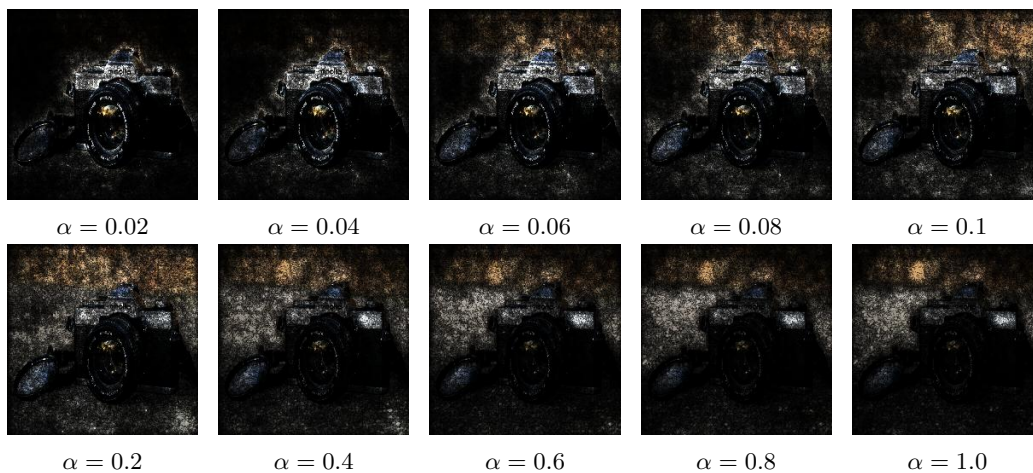

$\alpha = 0.02$ $\alpha = 0.04$ $\alpha = 0.06$ $\alpha = 0.08$ $\alpha = 0.1$

$\alpha = 0.2$ $\alpha = 0.4$ $\alpha = 0.6$ $\alpha = 0.8$ $\alpha = 1.0$

Figure 3: **Visualization of interior gradients.** Notice that the visualizations at lower values of the scaling parameter ($\alpha$) are sharper and much better at surfacing important features of the input image.

## 2.4 CUMULATING INTERIOR GRADIENTS

A different summarization of the interior gradients can be obtained by cumulating them. While there are a few ways of cumulating counterfactual gradients, the approach we take has the nice *attribution* property (Proposition 1) that the feature importance scores approximately add up to the prediction score. The feature importance scores are thus also referred to as *attributions*.

Notice that the set of counterfactual images $\{\alpha \text{ img} \mid 0 \leq \alpha \leq 1\}$ fall on a straight line path in $\mathsf{R}^{224 \times 224 \times 3}$. Interior gradients — which are the gradients of these counterfactual images — can be cumulated by integrating them along this line. We call the resulting gradients as **integrated gradients**. In what follows, we formalize integrated gradients for an arbitrary function $F : \mathsf{R}^n \rightarrow [0, 1]$ (representing a deep network), and an arbitrary set of counterfactual inputs falling on a path in $\mathsf{R}^n$.

Let $x \in \mathsf{R}^n$ be the input at hand, and $\gamma = (\gamma_1, \ldots, \gamma_n) : [0, 1] \to \mathsf{R}^n$ be a smooth function specifying the set of counterfactuals; here, $\gamma(0)$ is the *baseline* input (for Inception, a black image), and $\gamma(1)$ is the actual input (for Inception, the image being studied). Specifically, $\{\gamma(\alpha) \mid 0 \le \alpha \le 1\}$ is the set of counterfactuals (for Inception, a series of images that interpolate between the black image and the actual input).

The integrated gradient along the $i^{th}$ dimension for an input $x \in \mathsf{R}^n$ is defined as follows.

$$\mathsf{IntegratedGrads}_i(x) ::= \int_{\alpha=0}^{1} \frac{\partial F(\gamma(\alpha))}{\partial \gamma_i(\alpha)} \; \frac{\partial \gamma_i(\alpha)}{\partial \alpha} \; d\alpha \tag{5}$$

where $\frac{\partial F(x)}{\partial x_i}$ is the gradient of $F$ along the $i^{th}$ dimension at $x$.

A nice technical property of the integrated gradients is that they add up to the difference between the output of $F$ at the final counterfactual $\gamma(1)$ and the *baseline* counterfactual $\gamma(0)$. This is formalized by the proposition below, which is an instantiation of the fundamental theorem of calculus for path integrals.

**Proposition 1** *If $F : \mathsf{R}^n \to \mathsf{R}$ is differentiable almost everywhere* [2]*, and $\gamma : [0, 1] \to \mathsf{R}^n$ is smooth then*

$$\Sigma_{i=1}^n \mathsf{IntegratedGrads}_i(x) = F(\gamma(1)) - F(\gamma(0))$$

For most deep networks, it is possible to choose counterfactuals such that the prediction at the baseline counterfactual is near zero ($F(\gamma(0)) \approx 0$).[3] For instance, for the Inception network, the counterfactual defined by the scaling path satisfies this property as $\mathsf{Incp}(0^{224 \times 224 \times 3}) \approx 0$. In such cases, it follows from the Proposition that the integrated gradients form an attribution of the prediction output $F(x)$, i.e., they almost exactly distribute the output to the individual input features.

The additivity property provides a form of sanity checking for the integrated gradients and ensures that we do not under or over attribute to features. This is a common pitfall for attribution schemes based on feature ablations, wherein, an ablation may lead to small or a large change in the prediction score depending on whether the ablated feature interacts disjunctively or conjunctively to the rest of the features. This additivity is even more desirable when the networks score is numerically critical, i.e., the score is not used purely in an ordinal sense. In this case, the attributions (together with additivity) guarantee that the attributions are in the *units* of the score, and account for all of the score.

We note that these path integrals of gradients have been used to perform attribution in the context of small non-linear polynomials (Sun & Sundararajan (2011)), and also within the cost-sharing literature in economics where function at hand is a cost function that models the cost of a project as a function of the demands of various participants, and the attributions correspond to cost-shares. The specific path we use corresponds to a cost-sharing method called Aumann-Shapley (Aumann & Shapley (1974)).

**Computing integrated gradients.** The integrated gradients can be efficiently approximated by Riemann sum, wherein, we simply sum the gradients at points occurring at sufficiently small intervals along the path of counterfactuals.

$$\mathsf{IntegratedGrads}_i^{approx}(x) ::= \Sigma_{k=1}^m \frac{\partial F(\gamma(k/m))}{\partial \gamma_i(\alpha)} \; (\gamma(\tfrac{k}{m}) - \gamma(\tfrac{k-1}{m})) \tag{6}$$

Here $m$ is the number of steps in the Riemman approximation of the integral. Notice that the approximation simply involves computing the gradient in a for loop; computing the gradient is central to deep learning and is a pretty efficient operation. The implementation should therefore

---

[2] Formally, this means that the partial derivative of $F$ along each input dimension satisfies Lebesgue's integrability condition, i.e., the set of discontinuous points has measure zero. Deep networks built out of Sigmoids, ReLUs, and pooling operators should satisfy this condition.

[3] We did have trouble finding a baseline couterfactual for an RNN model that simulated the workings of a traffic light intersection between a main road and a side street; the naive benchmark counterfactual was one of no traffic at either intersection. But this did not have the lack of semantics that a black image or pure noise has for the Inception network. While no interesting labels are activated for the black image supplied to the Inception network, the same is not true for the "no traffic" benchmark supplied to the RNN model.

be straightforward in most deep learning frameworks. For instance, in TensorFlow (ten), it essentially amounts to calling `tf.gradients` in a loop over the set of counterfactual inputs (i.e., $\gamma(\frac{k}{m})$ for $k = 1, \ldots, m$), which could also be batched. Going forward, we abuse the term "integrated gradients" to refer to the approximation described above.

**Integrated gradients for Inception.** We compute the integrated gradients for the Inception network using the counterfactuals obtained by scaling the input image; $\gamma(\alpha) = \alpha$ img where img is the input image. Similar to the interior gradients, the integrated gradients can also be aggregated along the color channel to obtain pixel importance scores which can then be visualized as discussed earlier. Figure 4 shows these visualizations for a bunch of images. For comparison, it also presents the corresponding visualization obtained from the gradients at the actual image. From the visualizations, it seems quite evident that the integrated gradients are better at capturing important features.

## 2.5 TWO NORMATIVE AXIOMS FOR FEATURE ATTRIBUTIONS

We discuss two desirable axioms for feature attribution methods. We show that our integrated gradients method satisfies both. On the other hand, the other feature attribution methods in literature break one of the two axioms. These methods include DeepLift (Shrikumar et al. (2016)), Layer-wise relevance propagation (LRP) (Binder et al. (2016)), Deconvolutional networks (Zeiler & Fergus (2014)), and Guided back-propagation (Springenberg et al. (2014)).

**Sensitivity.**

A highly desirable property for feature attributions is *Sensitivity*. If a non-zero change in a single input variable (holding all other variables fixed) changes the output by a non-zero amount, then this variable should be given a non-zero attribution. In other words, attribution should be sensitive to change.

Integrated Gradients (ignoring the approximation in computing integrals) satisfies Sensitivity. The attribution to the variable is in fact equal to the change in function value (this is a one-variable instance of Proposition 1).

While this property is satisfied by the integrated gradients method, it is broken by Gradients, Deconvolution networks (DeConvNets) (Zeiler & Fergus (2014)), Guided back-propagation (Springenberg et al. (2014)).

Gradients break Sensitivity due to saturation (see Section 2.2), i.e., the prediction function may flatten at the input and thus have zero gradient despite the function value at the input being different from that at the benchmark. For a concrete example, consider a one variable, one ReLU network, $f(x) = 1 - \text{ReLU}(1 - x)$. Suppose we change the input from $x = 0$ to $x = 2$. The function changes from 0 to 1, but because $f$ is flat at $x = 1$, the gradient method gives attribution of 0 to $x$, violating sensitivity. We defer the counterexamples for other methods to Appendix B.

**Implementation Invariance.** Two networks can be functionally equivalent, i.e., their outputs are equal for all inputs, despite having very different implementations. We would like our attribution method to satisfy *Implementation Invariance*, i.e., the attributions are always identical for two functionally equivalent networks. To motivate this, notice that attribution can be colloquially defined as distributing the blame (or credit) for the output to the input features. Such a definition does not refer to implementation details. Moreover, the common practice of machine learning tends to evaluate the models from an input-output point of view, where implementations are purely means to an end.

Attributions generated by integrated gradients (or gradients, or any function of the interior gradients) satisfy Implementation Invariance since they are based only on the gradients of the function represented by the network. On the other hand, this fundamental property is unfortunately broken for the DeepLift and LRP methods. Below, we describe intuition for why Implementation Invariance is broken by these methods; a concrete example is provided in Appendix B.

First, notice that gradients are invariant to implementation. In fact, the chain-rule for gradients $\frac{\partial f}{\partial g} = \frac{\partial f}{\partial h} \cdot \frac{\partial h}{\partial g}$ is essentially about implementation invariance. To see this, think of $g$ and $f$ as the input and output of a system. The gradient of output $f$ to input $g$ can be computed either directly by

$\frac{\partial f}{\partial g}$, ignoring the intermediate function $h$, or by invoking the chain rule via $h$. This is exactly how backpropagation works.

As previously discussed, gradients don't satisfy sensitivity, and are therefore unsuitable for attribution. Methods like DeepLift tackle this issue by introducing a benchmark, and in some sense try to compute "discrete gradients" instead of gradients. They use a backpropagation procedure for composing discrete gradients. Unfortunately, such approaches are problematic because chain rule does not hold for discrete gradients in general. Formally $\frac{f(x_1)-f(x_0)}{g(x_1)-g(x_0)} = \frac{f(x_1)-f(x_0)}{h(x_1)-h(x_0)} \cdot \frac{h(x_1)-h(x_0)}{g(x_1)-g(x_0)}$ does not hold, and therefore these methods fail to satisfy implementation invariance.

If an attribution method fails to satisfy Implementation Invariance, the attributions are potentially sensitive to unimportant aspects of the models. For instance, in the example in Section B, the network architecture has more degrees of freedom than needed for representing the function, and as a result there are two set of values for the network parameters that lead to the same function. The training procedure can converge at either set of values depending on the initializtion or for other reasons, but the underlying network function would remain the same. It is undesirable that attributions differ for such reasons.

## 2.6 A FULL AXIOMATIZATION

There are many methods that satisfy Implementation Invariance and Sensitivity. In this section we show that Integrated Gradients is not just one of them. It is in fact also the only method that satisfies an extended set of axioms. The additional axioms are reasonably natural but perhaps not as fundamental to attribution. As we shall see in the next section there does not seem to be a perfect empirical evaluation for attribution methods. We hope that these axioms provide a theoretical framework for evaluating attribution methods, which provide a good complement to empirical evaluations.

As discussed earlier Integrated Gradients corresponds to a method called Aumann-Shapley studied by economists in the context of cost-sharing. (The function at hand is a cost-function whose input variables are demands of different participants and attributions correspond to cost-shares.) Here is the list of axioms, borrowed from the cost-sharing literature Billera & Heath (1982); a longer discussion of the desirability of these axioms in the context of attribution can be found in Sun & Sundararajan (2011).

- *Dummy*: If the function implemented by the deep network does not depend on a variable, then the attribution to it is always zero.

- *Additivity*: For all inputs, the attributions for a function $f_1 + f_2$ is the sum of the attributions for the function $f_1$ and the function $f_2$.

- *Completeness*: The attributions add up to the difference between the function values at the input and the benchmark.

- *Scale Invariance*: Informally, if the inputs to two networks differ in the scale of one of the variables (say Farenheit and Celsius), but have the same output for corresponding (rescaled) inputs, then the attributions should be identical.

- *Proportional Attributions for Homogenous Variables*: If a function can be represented by the sum of the two variables, then the two variables should receive attributions proportional to their input values.

**Proposition 2** *Billera & Heath (1982) Integrated Gradients is the unique method that satisfies all of the axioms above.*

## 2.7 AN EMPIRICAL EVALUATION OF OUR APPROACH

We now discuss an emprical evaluation of integrated gradients as a measure of feature importance, using gradients as a benchmark.

**Pixel ablations.**   The first evaluation is based on a method by Samek et al. (2015). Here we ablate[4] the top $5000$ pixels ($10\%$ of the image) by importance score, and compute the score drop for the highest scoring object class. The ablation is performed $100$ pixels at a time, in a sequence of $50$ steps. At each perturbation step $k$ we measure the average drop in score up to step $k$. This quantity is referred to a *area over the perturbation curve* (AOPC) by Samek et al. (2015).

Figure 5 shows the AOPC curve with respect to the number of perturbation steps for integrated gradients and gradients at the image. AOPC values at each step represent the average over a dataset of $150$ randomly chosen images. It is clear that ablating the top pixels identified by integrated gradients leads to a larger score drop that those identified by gradients at the image.

Having said that, we note an important issue with the technique. The images resulting from pixel perturbation are often unnatural, and it could be that the scores drop simply because the network has never seen anything like it in training.

**Localization.**   The second evaluation is to consider images with human-drawn bounding boxes around objects, and compute the percentage of pixel attribution inside the bounding box. We use the 2012 ImageNet object localization challenge dataset to get a set of human-drawn bounding boxes. We run our evaluation on $100$ randomly chosen images satisfying the following properties — (1) the total size of the bounding box(es) is less than two thirds of the image size, and (2) ablating the bounding box significantly drops the prediction score for the object class. (1) is for ensuring that the boxes are not so large that the bulk of the attribution falls inside them by definition, and (2) is for ensuring that the boxed part of the image is indeed responsible for the prediction score for the image. We find that on $82$ images the integrated gradients technique leads to a higher fraction of the pixel attribution inside the box than gradients at the actual image. The average difference in the percentage pixel attribution inside the box for the two techniques is $8.4\%$.

While these results are promising, we note the following caveat. Integrated gradients are meant to capture pixel importance with respect to the prediction task. While for most objects, one would expect the pixel located on the object to be most important for the prediction, in some cases the context in which the object occurs may also contribute to the prediction. The cabbage butterfly image from Figure 4 is a good example of this where the pixels on the leaf are also surfaced by the integrated gradients.

**Eyeballing.**   Ultimately, it was hard to come up with a perfect evaluation technique. So we did spend a large amount of time applying and eyeballing the results of our technique to various networks— the ones presented in this paper, as well as some networks used within products. For the Inception network, we welcome you to eyeball more visualizations in Figure 11 in the appendix and also at: `https://github.com/ankurtaly/Attributions`. While we found our method to beat gradients at the image for the most part, this is clearly a subjective process prone to interpretation and cherry-picking, but is also ultimately the measure of the utility of the approach—debugging inherently involves the human.

Finally, also note that we did not compare against other whitebox attribution techniques (e.g., DeepLift (Shrikumar et al. (2016))), because our focus was on black-box techniques that are easy to implement, so comparing against gradients seems like a fair comparison.

## 2.8   DEBUGGING NETWORKS

Despite the widespread application of deep neural networks to problems in science and technology, their internal workings largely remain a black box. As a result, humans have a limited ability to understand the predictions made by these networks. This is viewed as hindrance in scenarios where the bar for precision is high, e.g., medical diagnosis, obstacle detection for robots, etc. (dar (2016)). Quantifying feature importance for individual predictions is a first step towards understanding the behavior of the network; at the very least, it helps debug misclassified inputs, and sanity check the internal workings. We present evidence to support this below.

---

[4]Ablation in our setting amounts to zeroing out (or blacking out) the intensity for the R, G, B channels. We view this as a natural mechanism for removing the information carried by the pixel (than, say, randomizing the pixel's intensity as proposed by Samek et al. (2015), especially since the black image is a natural baseline for vision tasks.

We use feature importance to debug misclassifications made by the Inception network. In particular, we consider images from the ImageNet dataset where the groundtruth label for the image not in the top five labels predicted by the Inception network. We use interior gradients to compute pixel importance scores for both the Inception label and the groundtruth label, and visualize them to gain insight into the cause for misclassification.

Figure 6 shows the visualizations for two misclassified images. The top image genuinely has two objects, one corresponding to the groundtruth label and other corresponding to the Inception label. We find that the interior gradients for each label are able to emphasize the corresponding objects. Therefore, we suspect that the misclassification is in the ranking logic for the labels rather than the recognition logic for each label. For the bottom image, we observe that the interior gradients are largely similar. Moreover, the cricket gets emphasized by the interior gradients for the mantis (Inception label). Thus, we suspect this to be a more serious misclassification, stemming from the recognition logic for the mantis.

## 2.9 DISCUSSION

**Faithfullness.** A natural question is to ask why gradients of counterfactuals obtained by scaling the input capture feature importance for the original image. First, from studying the visualizations in Figure 4, the results look reasonable in that the highlighted pixels capture features representative of the predicted class as a human would perceive them. Second, we confirmed that the network too seems to find these features representative by performing ablations. It is somewhat natural to expect that the Inception network is robust to to changes in input intensity; presumably there are some low brightness images in the training set.

However, these counterfactuals seem reasonable even for networks where such scaling does not correspond to a natural concept like intensity, and when the counterfactuals fall outside the training set; for instance in the case of the ligand-based virtual screening network (see Section 3.1). We *speculate* that the reason why these counterfactuals make sense is because the network is built by composing ReLUs. As one scales the input starting from a suitable baseline, various neurons activate, and the scaling process that does a somewhat thorough job of exploring all these events that contribute to the prediction for the input. There is an analogous argument for other operator such as max pool, average pool, and softmax—here the triggering events arent discrete but the argument is analogous.

**Limitations of Approach.** We discuss some limitations of our technique; in a sense these are limitations of the problem statement and apply equally to other techniques that attribute to base input features.

- **Inability to capture Feature interactions:** The models could perform logic that effectively combines features via a conjunction or an implication-like operations; for instance, it could be that a molecule binds to a site if it has a certain structure that is essentially a conjunction of certain atoms and certain bonds between them. Attributions or importance scores have no way to represent these interactions.

- **Feature correlations:** Feature correlations are a bane to the understandability of all machine learning models. If there are two features that frequently co-occur, the model is free to assign weight to either or both features. The attributions would then respect this weight assignment. But, it could be that the specific weight assignment chosen by the model is not human-intelligible. Though there have been approaches to feature selection that reduce feature correlations (Yu & Liu (2003)), it is unclear how they apply to deep models on dense input.

## 2.10 RELATED WORK

Over the last few years, there has been a vast amount work on demystifying the inner workings of deep networks. Most of this work has been on networks trained on computer vision tasks, and deals with understanding what a specific neuron computes (Erhan et al. (2009); Le (2013)) and interpreting the representations captured by neurons during a prediction (Mahendran & Vedaldi (2015); Dosovitskiy & Brox (2015); Yosinski et al. (2015)).

Our work instead focuses on understanding the network's behavior on a specific input in terms of the base level input features. Our technique quantifies the importance of each feature in the prediction. Known approaches for accomplishing this can be divided into three categories.

**Gradient based methods.** The first approach is to use gradients of the input features to quantify feature importance (Baehrens et al. (2010); Simonyan et al. (2013)). This approach is the easiest to implement. However, as discussed earlier, naively using the gradients at the actual input does not accurate quantify feature importance as gradients suffer from saturation.

**Score back-propagation based methods.** The second set of approaches involve back-propagating the final prediction score through each layer of the network down to the individual features. These include DeepLift (Shrikumar et al. (2016)), Layer-wise relevance propagation (LRP) (Binder et al. (2016)), Deconvolutional networks (DeConvNets) (Zeiler & Fergus (2014)), and Guided back-propagation (Springenberg et al. (2014)). These methods largely differ in the backpropagation logic for various non-linear activation functions. While DeConvNets, Guided back-propagation and LRP rely on the local gradients at each non-linear activation function, DeepLift relies on the deviation in the neuron's activation from a certain baseline input.

Similar to integrated gradients, the DeepLift and LRP also result in an exact distribution of the prediction score to the input features. However, as shown by Figure 14, the attributions are not invariant across functionally equivalent networks. Besides, the primary advantage of our method over all these methods is its ease of implementation. The aforesaid methods require knowledge of the network architecture and the internal neuron activations for the input, and involve implementing a somewhat complicated back-propagation logic. On the other hand, our method is agnostic to the network architectures and relies only on computing gradients which can done efficiently in most deep learning frameworks.

**Model approximation based methods.** The third approach, proposed first by Ribeiro et al. (2016a;b), is to locally approximate the behavior of the network in the vicinity of the input being explained with a simpler, more interpretable model. An appealing aspect of this approach is that it is completely agnostic to the structure of the network and only deals with its input-output behavior. The approximation is learned by sampling the network's output in the vicinity of the input at hand. In this sense, it is similar to our approach of using counterfactuals. Since the counterfactuals are chosen somewhat arbitrarily, and the approximation is based purely on the network's output at the counterfactuals, the faithfullness question is far more crucial in this setting. The method is also expensive to implement as it requires training a new model locally around the input being explained.

## 3 APPLICATIONS TO OTHER NETWORKS

The technique of quantifying feature importance by inspecting gradients of counterfactual inputs is generally applicable across deep networks. While for networks performing vision tasks, the counterfactual inputs are obtained by scaling pixel intensities, for other networks they may be obtained by scaling an embedding representation of the input.

As a proof of concept, we apply the technique to the molecular graph convolutions network of Kearnes et al. (2016) for ligand-based virtual screening and an LSTM model (Zaremba et al. (2014)) for the language modeling of the Penn Treebank dataset (Marcus et al. (1993)).

### 3.1 LIGAND-BASED VIRTUAL SCREENING

The Ligand-Based Virtual Screening problem is to predict whether an input molecule is active against a certain target (e.g., protein or enzyme). The process is meant to aid the discovery of new drug molecules. Deep networks built using molecular graph convolutions have recently been proposed by Kearnes et al. (2016) for solving this problem.

Once a molecule has been identified as active against a target, the next step for medicinal chemists is to identify the molecular features—formally, *pharmacophores*[5]—that are responsible for the ac-

---

[5]A pharmacophore is the ensemble of steric and electronic features that is necessary to ensure the a molecule is active against a specific biological target to trigger (or to block) its biological response.

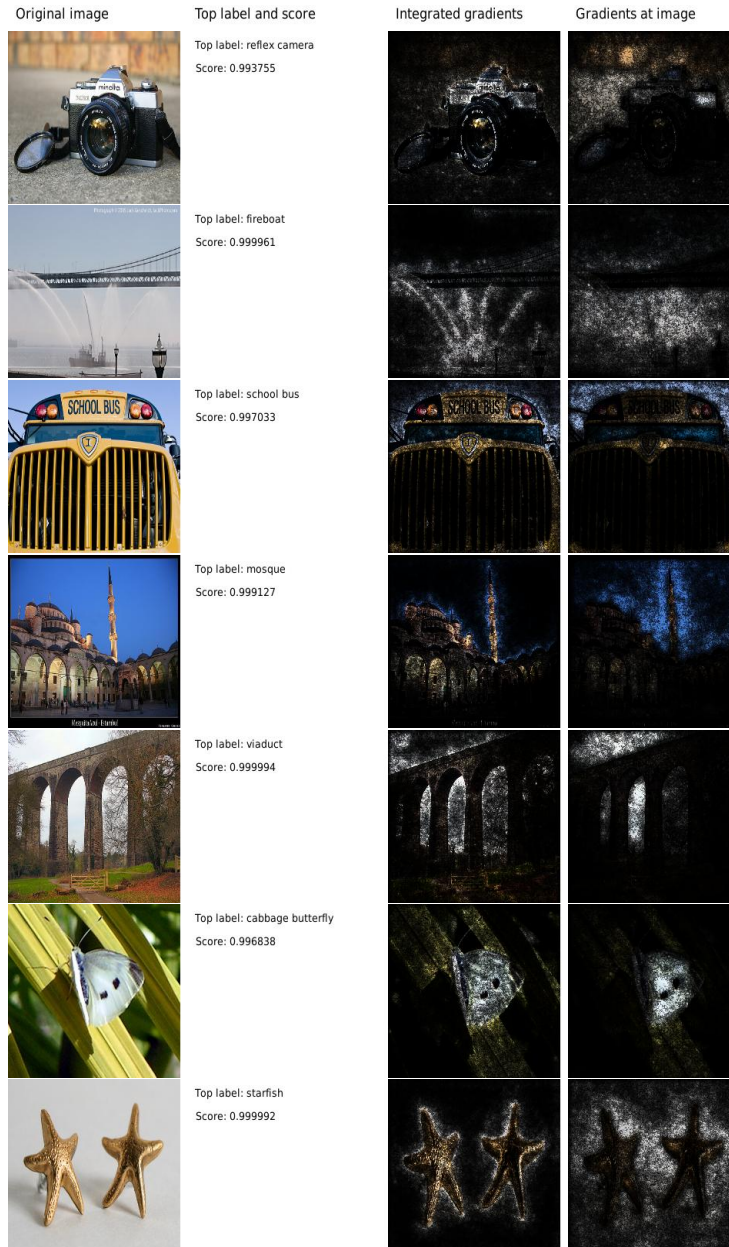

Figure 4: **Comparing integrated gradients with gradients at the image.** Left-to-right: original input image, label and softmax score for the highest scoring class, visualization of integrated gradients, visualization of gradients at the image. Notice that the visualizations obtained from integrated gradients are better at reflecting distinctive features of the image.

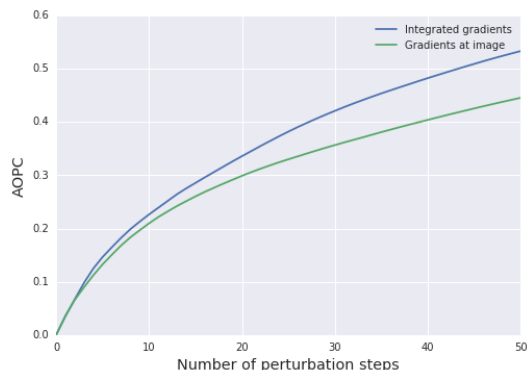

Figure 5: AOPC (Samek et al. (2015)) for integrated gradients and gradients at image.

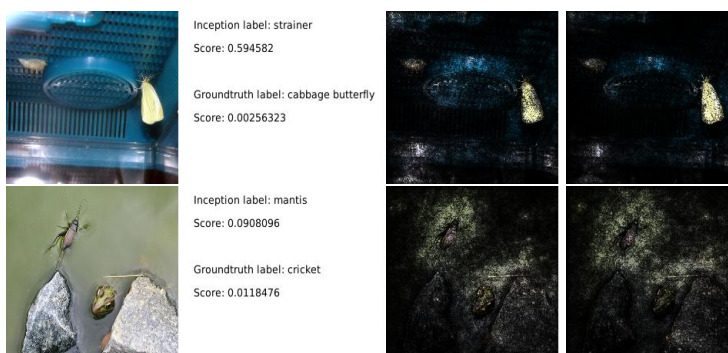

Figure 6: **Interior gradients of misclassified images.** Left-to-right: Original image, Softmax score for the top label assigned by the Inception network and the groundtruth label provided by ImageNet, visualization of integrated gradients w.r.t. Inception label, visualization of integrated gradients w.r.t. groundtruth label.

tivity. This is akin to quantifying feature importance, and can be achieved using the method of integrated gradients. The attributions obtained from the method help with identifying the dominant molecular features, and also help sanity check the behavior of the network by shedding light on its inner workings. With regard to the latter, we discuss an anecdote later in this section on how attributions surfaced an anomaly in W1N2 network architecture proposed by Kearns et al. (2016).

**Defining the counterfactual inputs.** The first step in computing integrated gradients is to define the set of counterfactual inputs. The network requires an input molecule to be encoded by hand as a set of atom and atom-pair features describing the molecule as an undirected graph. Atoms are featurized using a one-hot encoding specifying the atom type (e.g., C, O, S, etc.), and atom-pairs are featurized by specifying either the type of bond (e.g., single, double, triple, etc.) between the atoms, or the graph distance between them [6]

The counterfactual inputs are obtained by scaling down the molecule features down to zero vectors, i.e., the set $\{\alpha\text{Features(mol)} \mid 0 \leq \alpha \leq 1\}$ where $\text{Features}(mol)$ is an encoding of the molecule into atom and atom-pair features.

The careful reader might notice that these counterfactual inputs are not valid featurizations of molecules. However, we argue that they are still valid inputs for the network. First, all operators in the network (e.g., ReLUs, Linear filters, etc.) treat their inputs as continuous real numbers rather than discrete zeros and ones. Second, all fields of the counterfactual inputs are bounded between zero and one, therefore, we don't expect them to appear spurious to the network. We discuss this further in section 2.9

---

[6]This featurization is referred to as "simple" input featurization in Kearns et al. (2016).

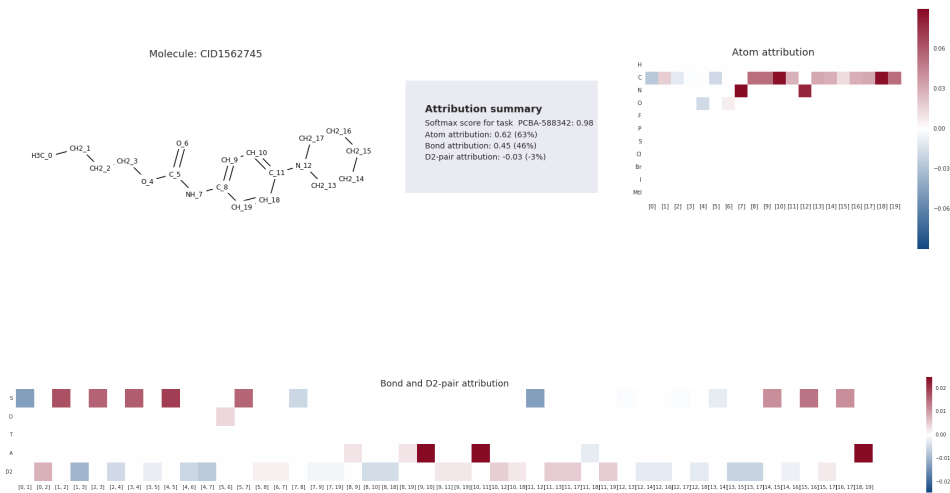

Figure 7: **Attribution for a molecule under the W2N2 network (Kearnes et al. (2016))**. The molecules is active on task PCBA-58432.

In what follows, we discuss the behavior of a network based on the W2N2-simple architecture proposed by Kearnes et al. (2016). On inspecting the behavior of the network over counterfactual inputs, we observe saturation here as well. Figure 13a shows the trend in the softmax score for the task PCBA-588342 for twenty five active molecules as we vary the scaling parameter $\alpha$ from zero to one. While the overall saturated region is small, saturation does exist near vicinity of the input $(0.9 \leq \alpha \leq 1)$. Figure 13b in the appendix shows that the total feature gradient varies significantly along the scaling path; thus, just the gradients at the molecule is fully indicative of the behavior of the network.

**Visualizing integrated gradients.** We cumulate the gradients of these counterfactual inputs to obtain an attribution of the prediction score to each atom and atom-pair feature. Unlike image inputs, which have dense features, the set of input features for molecules are sparse. Consequently, the attributions are sparse and can be inspected directly. Figure 7 shows heatmaps for the atom and atom-pair attributions for a specific molecule.

Using the attributions, one can easily identify the atoms and atom-pairs that that have a strongly positive or strongly negative contribution. Since the attributions add up to the final prediction score (see Proposition 1), the attribution magnitudes can be use for accounting the contributions of each feature. For instance, the atom-pairs that have a bond between them contribute cumulatively contribute 46% of the prediction score, while all other atom pairs cumulatively contribute $-3\%$.

We presented the attributions for 100 molecules active against a specific task to a few chemists. The chemists were able to immediately spot dominant functional groups (e.g., aromatic rings) being surfaced by the attributions. A next step could be cluster the aggregate the attributions across a large set of molecules active against a specific task to identify a common denominator of features shared by all active molecules.

**Identifying Dead Features.** We now discuss how attributions helped us spot an anomaly in the W1N2 architecture. On applying the integrated gradients method to the W1N2 network, we found that several atoms in the same molecule received the exact same attribution. For instance, for the molecule in Figure 7, we found that several carbon atoms at positions 2, 3, 14, 15, and 16 received the same attribution of $0.0043$ despite being bonded to different atoms, for e.g., Carbon at position 3 is bonded to an Oxygen whereas Carbon at position 2 is not. This is surprising as one would expect two atoms with different neighborhoods to be treated differently by the network.

On investigating the problem further we found that since the W1N2 network had only one convolution layer, the atoms and atom-pair features were not fully convolved. This caused all atoms that

have the same atom type, and same number of bonds of each type to contribute identically to the network. This is not the case for networks that have two or more convolutional layers.

Despite the aforementioned problem, the W1N2 network had good predictive accuracy. One hypothesis for this is that the atom type and their neighborhoods are tightly correlated; for instance an outgoing double bond from a Carbon is always to another Carbon or Oxygen atom. As a result, given the atom type, an explicit encoding of the neighborhood is not needed by the network. This also suggests that equivalent predictive accuracy can be achieved using a simpler "bag of atoms" type model.

## 3.2 LANGUAGE MODELING

To apply our technique for language modeling, we study word-level language modeling of the Penn Treebank dataset (Marcus et al. (1993)), and apply an LSTM-based sequence model based on Zaremba et al. (2014). For such a network, given a sequence of input words, and the softmax prediction for the next word, we want to identify the importance of the preceding words for the score.

As in the case of the Inception model, we observe saturation in this LSTM network. To describe the setup, we choose 20 randomly chosen sections of the test data, and for each of them inspect the prediction score of the next word using the first 10 words. Then we give each of the 10 input words a weight of $\alpha \in [0, 1]$, which is applied to scale their embedding vectors. In Figure 8, we plot the prediction score as a function of $\alpha$. For all except one curves, the curve starts near zero at $\alpha = 0$, moves around in the middle, stabilizes, and turns flat around $\alpha = 1$. For the interesting special case where softmax score is non-zero at $\alpha = 0$, it turns out that that the word being predicted represents out of vocabulary words. [!h]

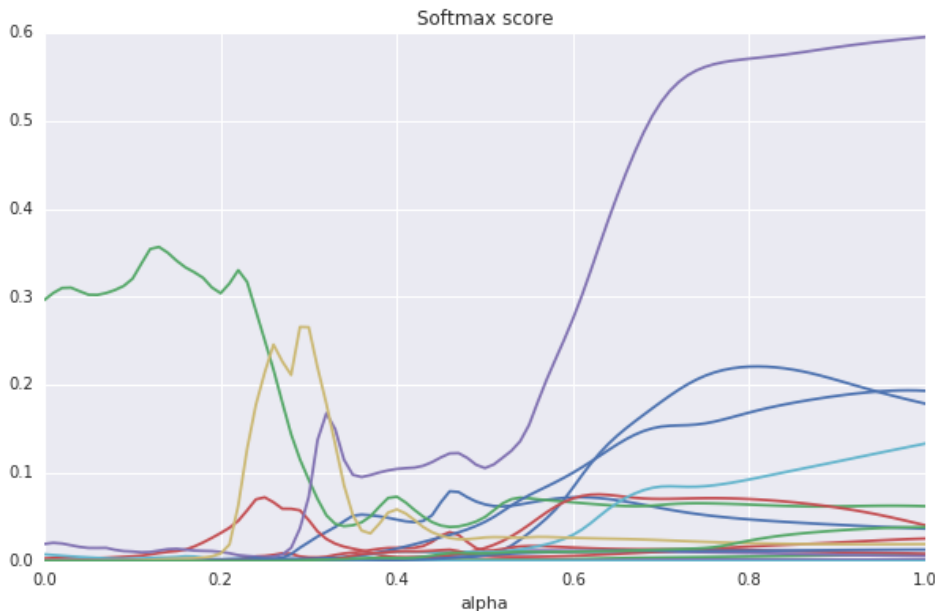

Figure 8: Softmax scores of the next word in the LSTM language model (Section 3.2)

In Table 9 and Table 10 we show two comparisons of gradients to integrated gradients. Due to saturation, the magnitudes of gradients are so small compared to the prediction scores that it is difficult to make sense of them. In comparison, (approximate) integrated gradients have a total amount close to the prediction, and seem to make sense. For example, in the first example, the integrated gradients attribute the prediction score of "than" to the preceding word "more". This makes sense as "than" often follows right after "more" in English. On the other hand, standard gradient gives a slightly negative attribution that betrays our intuition. In the second example, in predicting the second "ual", integrated gradients are clearly the highest for the first occurrence of

| Sentence | the | shareholders | claimed | more | **than** | $ N millions in losses |
|---|---|---|---|---|---|---|
| Integrated gradients | -0.1814 | -0.1363 | 0.1890 | **0.6609** | | |
| Gradients | 0.0007 | -0.0021 | 0.0054 | -0.0009 | | |

Figure 9: Prediction for **than**: 0.5307, total integrated gradient: 0.5322

| Sentence | and | N | minutes | after | the | ual | trading |
|---|---|---|---|---|---|---|---|
| Integrated gradients (*1e-3) | 0.0707 | 0.1286 | 0.3619 | 1.9796 | -0.0063 | **4.1565** | 0.2213 |
| Gradients (*1e-3) | 0.0066 | 0.0009 | 0.0075 | 0.0678 | 0.0033 | 0.0474 | 0.0184 |
| Sentence (Cont.) | halt | came | news | that | the | **ual** | group |
| Integrated gradients (*1e-3) | -0.8501 | -0.4271 | 0.4401 | -0.0919 | 0.3042 | | |
| Gradients (*1e-3) | -0.0590 | -0.0059 | 0.0511 | 0.0041 | 0.0349 | | |

Figure 10: Prediction for **ual**: 0.0062, total integrated gradient: 0.0063

"ual", which is the only word that is highly predictive of the second "ual". On the other hand, standard gradients are not only tiny, but also similar in magnitude for multiple words.

## 4 CONCLUSION

We present Interior Gradients, a method for quantifying feature importance. The method can be applied to a variety of deep networks without instrumenting the network, in fact, the amount of code required is fairly tiny. We demonstrate that it is possible to have some understanding of the performance of the network without a detailed understanding of its implementation, opening up the possibility of easy and wide application, and lowering the bar on the effort needed to debug deep networks.

We also wonder if Interior Gradients are useful within training as a measure against saturation, or indeed in other places that gradients are used.

ACKNOWLEDGMENTS

We would like to thank Patrick Riley and Christian Szegedy for their helpful feedback on the technique and on drafts of this paper.

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

# A APPENDIX

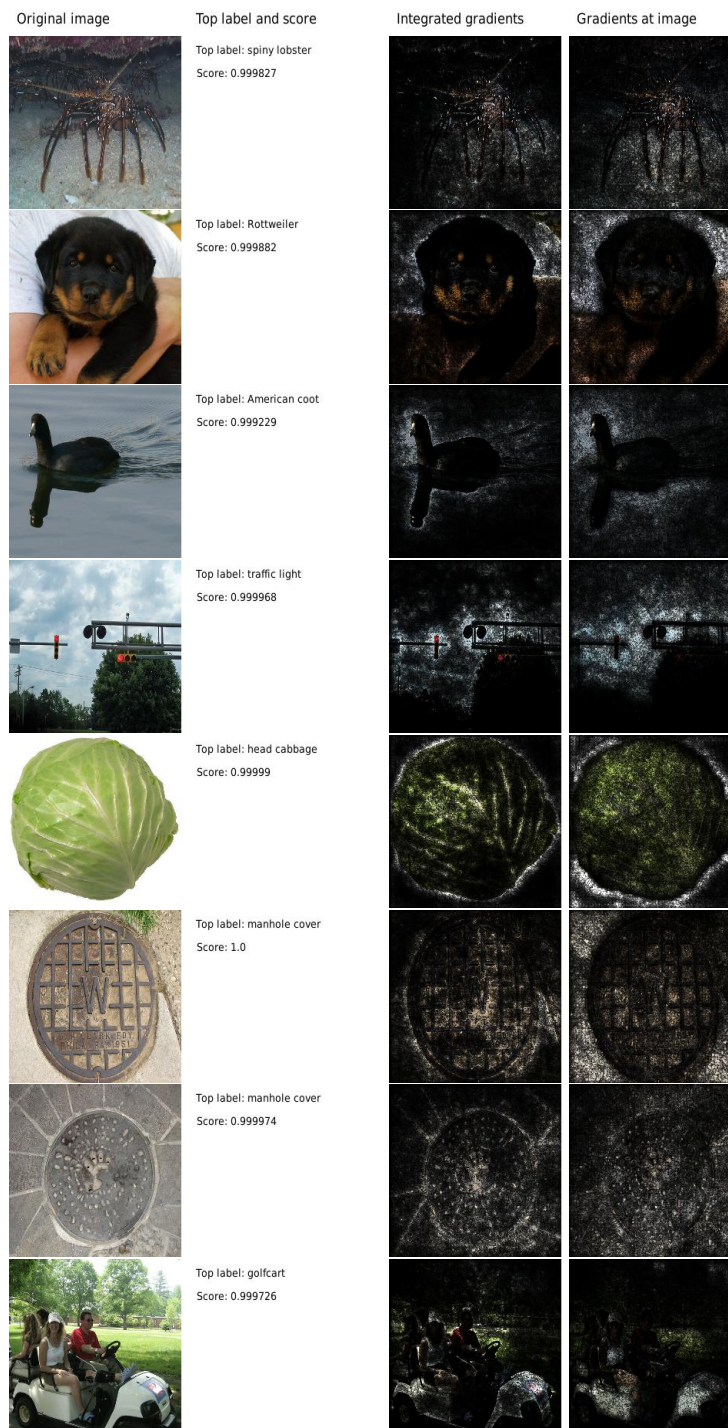

Figure 11: **More visualizations comparing integrated gradients with gradients at the image.**
Left-to-right: original input image, label and softmax score for the highest scoring class, visualization of integrated gradients, visualization of gradients at the image.

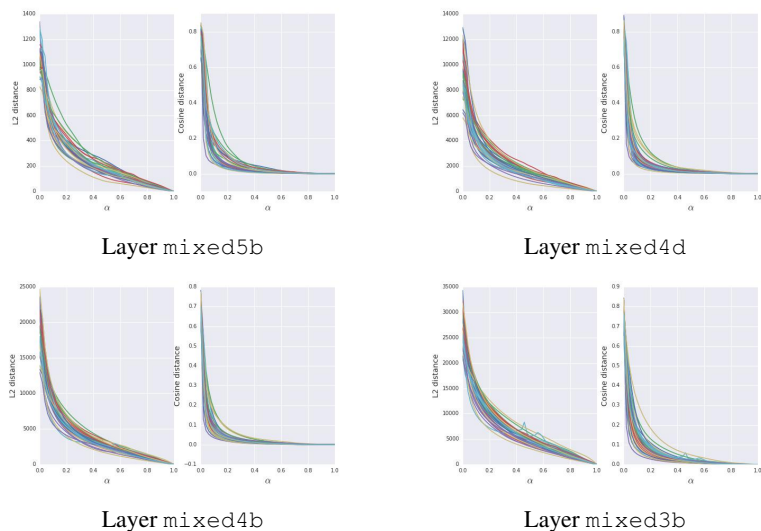

Figure 12: **Saturation in intermediate layers of Inception.** For each layer we plot the L2 and Cosine distance between the activation vector for a scaled down image and the actual input image, with respect to the scaling parameter. Each plot shows the trend for 30 randomly chosen images from the ImageNet dataset. Notice that trends in all plots flatten as the scaling parameter increases. For the deepest Inception layer mixed5b, the Cosine distance to the activation vector at the image is less than $0.01$ when $\alpha > 0.6$, which is really tiny given that this layer has 50176 neurons.

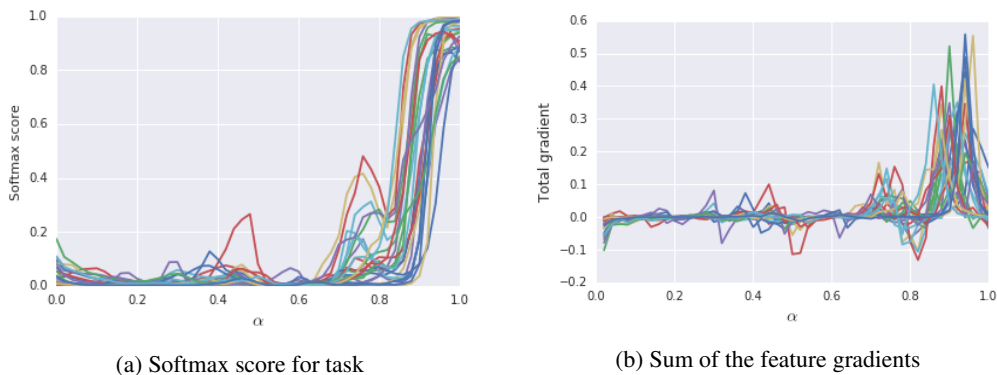

(a) Softmax score for task          (b) Sum of the feature gradients

Figure 13: **Saturation in the W2N2 network (Kearnes et al. (2016)).** Plots for the softmax score for task PCBA-58834, and the sum of the feature gradients w.r.t. the same task for twenty molecules. All molecules are active against the task

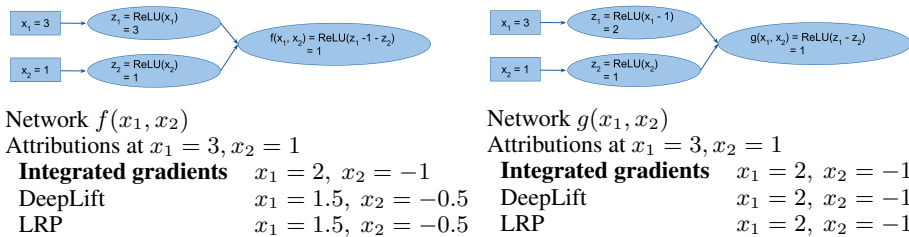

Network $f(x_1, x_2)$
Attributions at $x_1 = 3, x_2 = 1$

| | |
|---|---|
| **Integrated gradients** | $x_1 = 2,\ x_2 = -1$ |
| DeepLift | $x_1 = 1.5,\ x_2 = -0.5$ |
| LRP | $x_1 = 1.5,\ x_2 = -0.5$ |

Network $g(x_1, x_2)$
Attributions at $x_1 = 3, x_2 = 1$

| | |
|---|---|
| **Integrated gradients** | $x_1 = 2,\ x_2 = -1$ |
| DeepLift | $x_1 = 2,\ x_2 = -1$ |
| LRP | $x_1 = 2,\ x_2 = -1$ |

Figure 14: **Attributions for two functionally equivalent networks**. The figure shows attributions for two functionally equivalent networks $f(x_1, x_2)$ and $g(x_1, x_2)$ at the input $x_1 = 3$, $x_2 = 1$ using integrated gradients, DeepLift (Shrikumar et al. (2016)), and Layer-wise relevance propagation (LRP) (Binder et al. (2016)). The reference input for Integrated gradients and DeepLift is $x_1 = 0$, $x_2 = 0$. All methods except integrated gradients provide different attributions for the two networks.

## B    ATTRIBUTION COUNTER-EXAMPLES

We show that the methods DeepLift and Layer-wise relevance propagation (LRP) break the implementation invariance axiom, and the Deconvolution and Guided back-propagation methods break the sensitivity axiom.

Figure 14 provides an example of two equivalent networks $f(x_1, x_2)$ and $g(x_1, x_2)$ for which DeepLift and LRP yield different attributions.

First, observe that the networks $f$ and $g$ are of the form $f(x_1, x_2) = \mathsf{ReLU}(h(x_1, x_2))$ and $f(x_1, x_2) = \mathsf{ReLU}(k(x_1, x_2))^7$, where

$$h(x_1, x_2) = \mathsf{ReLU}(x_1) - 1 - \mathsf{ReLU}(x_2)$$
$$k(x_1, x_2) = \mathsf{ReLU}(x_1 - 1) - \mathsf{ReLU}(x_2)$$

Note that $h$ and $k$ are not equivalent. They have different values whenever $x_1 < 1$. But $f$ and $g$ are equivalent. To prove this, suppose for contradiction that $f$ and $g$ are different for some $x_1, x_2$. Then it must be the case that $\mathsf{ReLU}(x_1) - 1 \neq \mathsf{ReLU}(x_1 - 1)$. This happens only when $x_1 < 1$, which implies that $f(x_1, x_2) = g(x_1, x_2) = 0$.

Now we leverage the above example to show that Deconvolution and Guided back-propagation break sensitivity. Consider the network $f(x_1, x_2)$ from Figure 14. For a fixed value of $x_1$ greater than 1, the output decreases linearly as $x_2$ increases from 0 to $x_1 - 1$. Yet, for all inputs, Deconvolutional networks and Guided back-propagation results in zero attribution for $x_2$. This happens because for all inputs the back-propagated signal received at the node $\mathsf{ReLU}(x_2)$ is negative and is therefore not back-propagated through the $\mathsf{ReLU}$ operation (per the rules of deconvolution and guided back-propagation; see Springenberg et al. (2014) for details). As a result, the feature $x_2$ receives zero attribution despite the network's output being sensitive to it.

---

[7] $\mathsf{ReLU}(x)$ is defined as $\mathsf{max}(x, 0)$.

