# Peer review of "Gradients of Counterfactuals"

_ICLR 2017 — rejected_

[Official Review · AnonReviewer1 · rating 5 · confidence 4 · 16 Dec 2016]
**Scaling input samples.**

This work proposes to use visualization of gradients to further understand the importance of features (i.e. pixels) for visual classification. Overall, this presented visualizations are interesting, however, the approach is very ad hoc. The authors do not explain why visualizing regular gradients isn't correlated with the importance of features relevant to the given visual category and proceed to the interior gradient approach. 

One particular question with regular gradients at features that form the spatial support of the visual class. Is it the case that the gradients of the features that are confident of the prediction remain low, while those with high uncertainty will have strong gradients?

With regards to the interior gradients, it is unclear how the scaling parameter \alpha affects the feature importance and how it is related to attention.

Finally, does this model use batch normalization?

[Official Review · AnonReviewer3 · rating 3 · confidence 4 · 17 Dec 2016]

The authors propose to measure “feature importance”, or specifically, which pixels contribute most to a network’s classification of an image. A simple (albeit not particularly effective) heuristic for measuring feature importance is to measure the gradients of the predicted class wrt each pixel in an input image I. This assigns a score to each pixel in I (that ranks how much the output prediction would change if a given pixel were to change). In this paper, the authors build on this and propose to measure feature importance by computing gradients of the output wrt scaled version of the input image, alpha*I, where alpha is a scalar between 0 and 1, then summing across all values of alpha to obtain their feature importance score. Here the scaling is simply linear scaling of the pixel values (alpha=0 is all black image, alpha=1 is original image). The authors call these scaled images “counterfactuals” which seems like quite an unnecessarily grandiose name for literally, a scaled image. 

The authors show a number of visualizations that indicate that the proposed feature importance score is more reasonable than just looking at gradients only with respect to the original image. They also show some quantitative evidence that the pixels highlighted by the proposed measure are more likely to fall on the objects rather than spurious parts of the image (in particular, see figure 5). The method is also applied to other types of networks. The quantitative evidence is quite limited and most of the paper is spent on qualitative results.

While the goal of understanding deep networks is of key importance, it is not clear whether this paper really help elucidate much. The main interesting observation in this paper is that scaling an image by a small alpha (i.e. creating a faint image) places more “importance” on pixels on the object related to the correct class prediction. Beyond that, the paper builds a bit on this, but no deeper insight is gained. The authors propose some hand-wavy explanation of why using small alpha (faint image) may force the network to focus on the object, but the argument is not convincing. It would have been interesting to try to probe a bit deeper here, but that may not be easy.

Ultimately, it is not clear how the proposed scheme for feature importance ranking is useful. First, it is still quite noisy and does not truly help understand what a deep net is doing on a particular image. Performing a single gradient descent step on an image (or on the collection of scaled versions of the image) hardly begins to probe the internal workings of a network. Moreover, as the authors admit, the scheme makes the assumption that each pixel is independent, which is clearly false.

Considering the paper presents a very simple idea, it is far too long. The main paper is 14 pages, up to 19 with references and appendix. In general the writing is long-winded and overly verbose. It detracted substantially from the paper. The authors also define unnecessary terminology. “Gradients of Coutnerfactuals” sounds quite fancy, but is not very related to the ideas explored in the writing. I would encourage the authors to tighten up the writing and figures down to a more readable page length, and to more clearly spell out the ideas explored early on.

[Official Review · AnonReviewer2 · rating 3 · confidence 4 · 18 Dec 2016]
**review: lacking experimental comparison to prior work**

This paper proposes a new method, interior gradients, for analysing feature importance in deep neural networks.  The interior gradient is the gradient measured on a scaled version of the input.  The integrated gradient is the integral of interior gradients over all scaling factors.  Visualizations comparing integrated gradients with standard gradients on real images input to the Inception CNN show that integrated gradients correspond to an intuitive notion of feature importance.

While motivation and qualitative examples are appealing, the paper lacks both qualitative and quantitative comparison to prior work.  Only the baseline (simply the standard gradient) is presented as reference for qualitative comparison.  Yet, the paper cites numerous other works (DeepLift, layer-wise relevance propagation, guided backpropagation) that all attack the same problem of feature importance.  Lack of comparison to any of these methods is a major weakness of the paper.  I do not believe it is fit for publication without such comparisons.  My pre-review question articulated this same concern and has not been answered.

[Author Response · Ankur Taly · 22 Dec 2016]
**Response to AnonReviewer2 and AnonReviewer3**

We thank the reviewers for a detailed review.  The rebuttal below addresses some of the mentioned concerns.

Regarding “far too long” and “unnecessarily grandiose name for literally, a scaled image”: 

We’d agree that the paper is long for the ideas in it. The length stems from the difficulty of not having a crisp evaluation technique for feature importance. So we try to resort to qualitative discussions together with images. But we  can definitely try to tighten the writing. We are open to changing the title of the paper to “Interior Gradients” or something like it, though it is worth noting that while scaling intensities seems natural for images, analogous scaling for Text or Drug Discovery models results in inputs that are more obviously fake, i.e., counterfactual.

Regarding “how the proposed scheme for feature importance ranking is useful”: 

While debugging deep networks is hard in general, examining feature importance scores offers a limited but useful insight into the operation of the network on a particular input. For us, the experience with the Drug Discovery network where we found, via our attributions, that the bond features were severely underused (see Section 3.1) was a concrete instance of how feature importance analysis could help debug and improve networks. As we discussed in section 2.7, we do mention the limitations of our technique in understanding what the network does. The same pros and cons would seem to apply to other feature importance techniques (see Section 2.8). The key difference is that ours is much easier to implement--- as simple as computing a gradient.

Regarding “The quantitative evidence is quite limited and most of the paper is spent on qualitative results”: 

We address with the following multipart response; apologies for the lengthy response.

First, we do plan to produce a comparison with side by sides for LRP and our method for the MNIST data set over the next few weeks as a sanity check.

However, we don’t think that that there is a strong metric to compare different feature importance techniques. This is acknowledged by Samek et al in their 2015 ICML Visualization Workshop work. We elaborate on this further at the end of this rebuttal.  

Methods like DeepLift and Layer-wise Relevance Propagation (LRP) break a fundamental axiom in our mind: the attributions depend on the implementation, i.e. two networks that implement identical input-output  relationships can have different attributions. This seems odd---see Section 2.4 and Figure 14.  Perhaps, we did not emphasize this enough in the paper. 

The main focus for us in the evaluation conducted so far has been to ensure that our output was sensible. In Section 2.5, we discuss a combination of approaches that we used to assess the attributions, including eyeballing, localization, and ablations. We welcome you to visualize more attributions at:

[Author Response · Ankur Taly · 11 Jan 2017]
**New material added to the paper**

We added two new sections (2.5 and 2.6) to the paper. Section 2.5
proposes two very desirable axioms for attribution methods,
and uses them to rule out other attribution methods from the literature.
Section 2.6 proposes a full axiomatization under which our method is
unique.

It is possible that sections 2.3-2.7 may constitute a shorter 6-page
self-contained paper with the title --- "attributions using interior
gradients".

[Final Decision · Program Chairs · 06 Feb 2017]
**ICLR committee final decision**

This paper was reviewed by 3 experts. All 3 seem unconvinced of the contributions, point to several shortcomings, and recommend rejection. I see no basis for overturning their recommendation. To be clear, the problem of achieving insight into the inner workings of deep networks is of significant importance and I encourage the authors to use the feedback to improve the manuscript.